

# Predicting the potential distributions of the invasive cycad scale *Aulacaspis yasumatsui* (Hemiptera: Diaspididae) under different climate change scenarios and the implications for management

Jiufeng Wei[1,*], Qing Zhao[1,*], Wanqing Zhao[1] and Hufang Zhang[2]

[1] Department of Entomology, Shanxi Agricultural University, Taigu, Shanxi, P.R. China
[2] Department of Biology, Xinzhou Teachers University, Xinzhou, Shanxi, P.R. China
[*] These authors contributed equally to this work.

Corresponding author
Hufang Zhang, zh_hufang@sohu.com

## ABSTRACT

Cycads are an ancient group of gymnosperms that are popular as landscaping plants, though nearly all of them are threatened or endangered in the wild. The cycad aulacaspis scale (CAS), *Aulacaspis yasumatsui* Takagi (Hemiptera: Diaspididae), has become one of the most serious pests of cycads in recent years; however, the potential distribution range and the management approach for this pest are unclear. A potential risk map of cycad aulacaspis scale was created based on occurrence data under different climatic conditions and topology factors in this study. Furthermore, the future potential distributions of CAS were projected for the periods 2050s and 2070s under three different climate change scenarios (GFDL-CM3, HADGEM2-AO and MIROC5) described in the Special Report on Emissions Scenarios of the IPCC (Intergovernmental Panel on Climate Change). The model suggested high environmental suitability for the continents of Asia and North America, where the species has already been recorded. The potential distribution expansions or reductions were also predicted under different climate change conditions. Temperature of Driest Quarter (Bio9) was the most important factor, explaining 48.1% of the distribution of the species. The results also suggested that highly suitable habitat for CAS would exist in the study area if the mean temperature of 15–20 °C in the driest quarter and a mean temperature of 25–28 °C the wettest quarter. This research provides a theoretical reference framework for developing policy to manage and control this invasive pest.

## INTRODUCTION

Invasive species threaten native ecosystems through interspecific competition with native species, but they also threaten human-managed systems, such as those related to agriculture, animal health and forestry (*Bradley, Wilcove & Oppenheimer, 2000*; *Simberloff et al., 2003*). Considerable evidence has shown that climate change will aggravate the

impacts of the naturalization and subsequent invasion of invasive species into novel communities and ecosystems (*Walsh, Carpenter & Zanden, 2016*; *Ekesi et al., 2016*; *Paini et al., 2016*). Moreover, rising temperatures can significantly influence the physiological characteristics of insect lift history and population dynamics, such as development rate, voltinism and distribution range (*Bellard et al., 2013*; *Yousuf et al., 2014*; *Duan et al., 2014*). Understanding the change in potential distribution due to climate change is a critical foundation that is required to manage and control the introduction of alien species (*Massin et al., 2012*).

Cycads (Cycadales), commonly called "sago palms", are considered as the most primitive extant seed plant lineage and are often characterized as "living fossils" (*Marler & Moore, 2010*) due to their long evolutionary history (*Brenner, Stevenson & Twigg, 2003*; *Salas-Leiva et al., 2013*). There are 348 extant species of cycads, and almost all are threatened or endangered (*Normark et al., 2017*). At present, cycads are popular landscape plants in tropical and subtropical areas because they are long-lived and require little maintenance (*Bailey, Chang & Lai, 2011*). However, many cycad species, such as *Cycad micronesica*, are facing extinction in the wild due to insect pests (e.g., the cycad blue butterfly) (*Wu et al., 2010*) and microbial pathogens (*Nesamari et al., 2015*). The cycad aulacaspis scale (CAS), *Aulacaspis yasumatsui* Takagi (Hemiptera: Diaspididae), is one of the serious pest of cycads (*Giorgi & Vandenberg, 2012*). Studies have suggested that this highly invasive species poses a threat to costly ornamental and horticultural plants, but it also damages wild cycad populations, including plants in conservation areas around the world (*Marler & Lawrence, 2012*). The CAS was first described from specimens collected in Bangkok, Thailand in 1972 by *Takagi (1977)*, but the species usually occurs at low densities in its native range due to control by natural enemies (*Tang, Yang & Vatcharakorn, 1997*; *Marler & Moore, 2010*). To date, the CAS has spread to many areas outside of its native range. The first report of CAS on cycads outside Thailand was in 1996 in Florida, USA, where it infested ornamental plants and killed large numbers of king sago, *Cycas revoluta* Thunberg (*Howard et al., 1999*); it was subsequently found in the Fairchild Tropical Garden and the Montgomery Botanical Center, both of which have collections of rare and endangered cycads. Following its introduction in Florida, the CAS spread to Alabama, Georgia, Louisiana, South Carolina, and Texas (*Haynes, 2005*) and most recently, this pest was also reported in Mexico (*González-Gómez et al., 2016*; *Normark et al., 2017*). The CAS has now been introduced to many countries including China (*Bailey et al., 2010*), Singapore (*Hodgson & Martin, 2001*), Vietnam (*EPPO, 2017*) and Philippines in Asia; Guam (*Terry & Marler, 2005*), the Cayman Islands, Puerto Rico, the Vieques Islands, and the Hawaiian Islands in the Americas; as well as in Africa (*Nesamari et al., 2015*); and Europe (*Malumphy & Marquart, 2012*); among other locations. Moreover, CAS was intercepted several times from France in 2001 (*Germain, 2002*).

Due to the damage caused by CAS, many cycads in southern Miami, area of Florida, died within a year, and mature cycads were killed in a matter of months (*Walters, Shroyer & Anderson, 1997*; *Howard et al., 1999*). Some endemic cycads, such as *Cycas micronesica* in Guam (*Marler & Lawrence, 2012*) and *Cycas taitungensis* in Taiwan (*Bailey, Chang & Lai, 2011*) are threatened with extinction due to CAS. The pest feeds on both the aerial

parts of host cycads but also on their roots (*Takagi & De faveri, 2009*), and at the start of an infestation, damage initially appears as chlorotic spots on the undersides of leaves. As the infestation progresses, CAS infests the upper surfaces of the leaf and then occurs on the petioles as well as the stems and as the population density of the scale increases, infested plants become quickly and almost completely covered by live and dead scales and appear as enveloped a white crust (*Emshousen, Mannion & Glenn, 2004*). CAS is very harmful to cycads, causing death as well as reducing their ornamental value (*Milek, Šimala & Novak, 2008*).

Currently, at least eight genera from three different families of cycads are threatened by CAS (*García et al., 2017*), and there is an urgent need to control and manage this pest due to its rapid spread and wide host range. If this pest is not managed or controlled, it poses a threat to the ornamental cycad nursery industry and may accelerate the extinction of wild cycads (*Emshousen, Mannion & Glenn, 2004*).

Based on empirical data from Florida and other regions, it is difficult to control CAS through any of the currently available methods (*Muniappan et al., 2012*). Therefore, an alternative approach is to enforce strict quarantine measures in countries where CAS has not yet been introduced by prohibiting the importation of cycad plants from infested countries. Information regarding the potential distribution of this species under climate change will be indispensable for scientists and farmers around the world to develop future monitoring and management strategies for this pest (*Jiménez-Valverde et al., 2011*). In this paper, the current and future potential distributions of CAS were estimated based on available occurrence data using MaxEnt software. The mainly objectives were (1) to identify the main climatic variable that constrains the potential distribution of CAS, (2) to predict the trends of its suitable habitat range under different climatic change scenarios, and (3) to provide a theoretical reference framework for developing policy to manage and control this invasive pest.

## MATERIALS AND METHODS

### Species data

The occurrence data for CAS were compiled from three main sources: the Centre for Agriculture and Bioscience International (CABI, https://www.cabi.org/), the Global Biodiversity Information Facility (GBIF, https://www.gbif.org/) and the existing literature (Table S1). Geo-coordinates for each chosen data-point were either referenced from information in the literature or by using Google Earth coordinates. Duplicate or unclear locations were excluded, and data were manually checked for accuracy.

Occurrence records are often biased towards areas where assemblages are easily accessible, such as near cities or other areas with high population density (*Kadmon, Farber & Danin, 2004*); therefore, occurrence data can significantly impacts the modelling results of the model (*Elith et al., 2011*). To minimize sampling bias and remove spatial autocorrelation, a coarse resolution (5 km × 5 km) was created using the workflow of *Li, Du & Guo (2015)* and a single point was randomly selected from each cell containing one or more sampling points (*Rodriguez-Castaneda et al., 2012*). The workflow was conducted

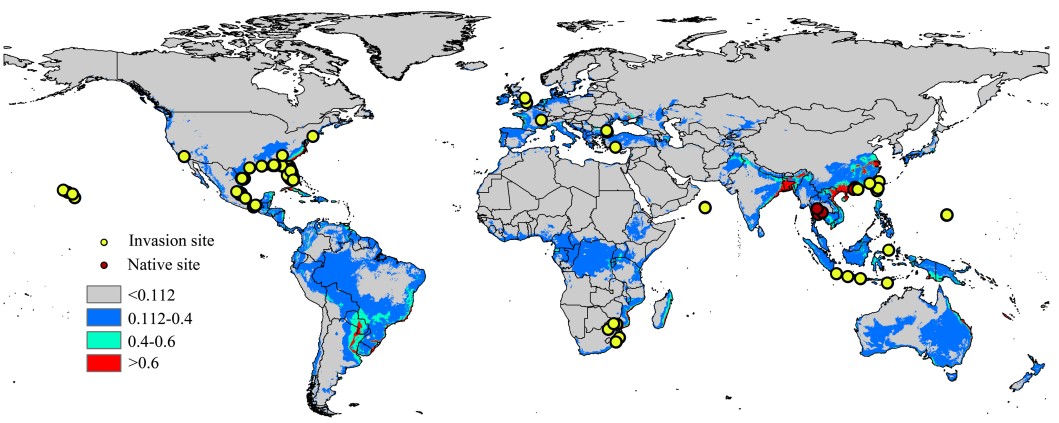

**Figure 1 Native and invasive localities of CAS used in current modeling.** Red dot represent native localities and green dot represent invasive localities. The base map was created with Natural Earth Dataset (http://www.naturalearthdata.com/). Gray, unsuitable habitat area; Blue, low habitat suitability area; Green, moderate habitat suitability area; Red, highly habitat suitability area.

in ArcGIS 10.1 (ESRI, Redlands, CA, USA. http://www.esri.com/), and after filtering, 94 locations remained, including the native range (6 points) and the invaded regions in North America (59 points), Europe (7 points), Asia (16 points) and Africa (6 points). The list of distribution sites is shown in Fig. 1 and Table S2.

## Environmental variables

The environmental variables used to characterize ecological niches were selected considering the climate and topography. Climatic variables are probably the main determinant of species niches at large scale (*Luoto, Virkkala & Heikkinen, 2007*), and they have previously been used for insect niche modelling (*Medley, 2010*). All environmental variables used in this study were download from the WorldClim Global Climate Database (http://www.worldclim.org) (*Hijmans et al., 2005*), and current climate conditions were represented by minima, maxima and average values of monthly, quarterly and annual ambient temperatures as well as precipitation values recorded from 1950 to 2000. These parameters provided a combination of means, extremes and seasonal differences in variables known to influence species distribution (*Root et al., 2003*). Topographic variables represented by altitude (ALT) were derived from HYDRO1K (http://eros.usgs.gov). Multicollinearity among predictor variables may hamper the analysis of species-environment relationships (*Heikkinen et al., 2006*). Thus, principal component analysis (PCA) in SPSS Statistics was performed to remove highly correlated variables ($r \geq |0.85|$) (Table S3). Finally, six climatic environmental and topographic variables were selected: Mean Diurnal Range (Mean of monthly (max temp–min temp)) (Bio2), Mean Temperature of Wettest Quarter (Bio8), Mean Temperature of Driest Quarter (Bio9), Precipitation Seasonality (Coefficient of Variation) (Bio15), Precipitation of Coldest Quarter (Bio19) and Altitude (ALT).

Given the uncertainty in scenarios of future climate scenarios, impact assessments should incorporate data from a range of climate models that effectively simulate historical

climate over the area of interest (*Guisan et al., 2013*). Therefore, to predict the future potential distribution of CAS on a global scale, the Hadley Global Environment Model 2-Atmosphere Ocean (HADGEM2-AO), Geophysical Fluid Dynamics Laboratory Coupled Model v3 (GFDL-CM3) and Model for Interdisciplinary Research on Climate (MIROC5) for 2050 (average for 2041–2060) and 2070 (average for 2061–2080) were obtained from the Coupled Model Intercomparison Project Phase 5 (CMIP5) of the fifth assessment of the International Panel on Climate Change (IPCC) in 2010 (*Moss et al., 2010*). Four possible greenhouse gas concentration trajectories were selected that ranged from low (RCP 2.6) to high (RCP 8.5) and corresponded to increases in global radiative forcing values in the year 2100 relative to preindustrial values (2.6, 4.5, 6.0 and 8.5 w/m$^2$, respectively). Additionally, it was assumed that the global mean temperature will increase by 0.3 °C–4.8 °C by 2100 across all RCPs (*Van Vuuren et al., 2011*). Four climate change scenario/year combinations were selected to simulate the future influence of potentially extreme climatic variations on the geographical distribution patterns: RCP 2.6-2050, RCP 8.5-2050, RCP 2.6-2070 and RCP 8.5-2070. All future climate projection data were download from the WorldClim Database (http://www.worldclim.org). All current and future data had a spatial resolution of 2.5 arc min (approx. ~5 km resolution at the equator).

## MaxEnt modelling

In recent years, MaxEnt software (MaxEnt version 3.3.3k, http://www.cs.princeton.edu/~schapire/maxent/) has frequently been used to simulate shifts in species ranges under current and future climate scenarios (*Ponce-Reyes et al., 2012*; *Wong et al., 2013*; *Cooper et al., 2016*). This software is popular because it requires only presence-only data. Comparative studies have consistently shown that MaxEnt has excellent performance and outperforms than many other methods (such as GARP) in estimating potential species distributions, particularly when sample sizes are small (*Elith et al., 2006*; *Bosso et al., 2016*; *Sultana et al., 2017*). Moreover, some studies found that this software performed well regardless of the number or geographical extent of species records, compared to the performance of mahalannobis typicalities, random forests methods and other methods (*Bosso et al., 2016*; *Hernandez et al., 2006*; *Elith et al., 2011*).

Building models from MaxEnt with an appropriate amount of complexity for the study objectives is critical for robust inference (*Merow et al., 2014*). Moreover, for several reasons, the default settings of MaxEnt are expected to produce overfit models (*Radosavijevic & Anderson, 2014*). Recent studies have demonstrated the importance of both balancing model complexity and predictive power and evaluating model's performance with spatially independent data. Hence, in order to produce the best possible model for CAS, avoiding overfitting while maximizing predictive power, we employed the R package ENMeval to select the optimal combination of two important MaxEnt's parameters, the value of the regularization multiplier and the combination of feature classes (*Muscarella et al., 2014*; *Warren & Seifert, 2011*). In this study, the "checkerboard2" approach was used by calculating the standardized Akaike information criterion coefficient (AICc), and the parameterizations that resulted in the model with the lowest delta AICc score were selected to run the final MaxEnt models. The regularization multiplier was varied from 0.5 to 4

in increments of 0.5, and the following six feature combinations were tested: (1) Linear (L); (2) Linear (L) and Quadratic (Q); (3) Hinge (H); (4) Linear (L), Quadratic (Q) and Hinge (H); (5) Linear (L), Quadratic (Q), Hinge (H) and Product (P); (6) Linear (L), Quadratic(Q), Hinge (H), Product (P) and Threshold (T). The ENMeval package were implemented in R 3.1.3 (*R Core Team, 2015*). The result were shown in Table S4 and Fig. S1. Thus, a regularization multiplier = 2.5; feature combinations = Linear, Quadratic and Hinge (LQH) were choose in final MaxEnt configuration. Background points were randomly chosen within the area enclosed by a minimum convex polygon comprising all records of the species of interest as suggested by *Phillips (2008)*. The logistic output of MaxEnt was used for all the analyses. The 10th percentile training presence logistic threshold was used to define the suitable and unsuitable habitats for CAS. This threshold is a conservative value that is widely adopted in species distribution modelling, especially when data have been collected by different observers and have been collected using different methods over a long time (*Bosso et al., 2016*). By applying this threshold, binary presence/absence maps were created using the reclassify module from ArcGIS 10.1 (ESRI, Redlands, CA, USA). To maximize the predictive information and simplification of future analysis, the suitable habitat areas for CAS were reclassified into four levels: unsuitable habitat (no risk), low habitat suitability (low risk), moderate habitat suitability (medium risk) and high habitat suitability (high risk). The area under the curve (AUC) of the receiver operating characteristic (ROC) (*Fielding & Bell, 1997*) and the true skill statistic (TSS; sensitivity + specificity − 1) (*Allouche, Tsoar & Kadmon, 2006*) were used to estimate the performance of the model according to *Swets (1988)*. The AUC value ranges from 0 to 1, where a value below 0.5 can be interpreted as a random prediction; 0.5–0.7 indicates poor model performance; 0.7–0.9 indicates moderate performance; and a value above 0.9 is considered to have "good" discrimination abilities (*Peterson et al., 2011*). The TSS accounts for both omission and commission errors, and the value range from −1 to 1, where 1 indicates perfect agreement and 0 represents a random fit. Excellent model performances are expressed by TSS values close to 1 (*Allouche, Tsoar & Kadmon, 2006*).

A 10-fold cross-validation was used to run MaxEnt to prevent random errors from affecting the selection of the validation and prediction sample. To assess the influence of environmental variables on species, the jackknife test was used to measure the importance and the percent contributions of each variable. To account for variation among GCMs, a final projection environmental suitability map was created for CAS by averaging the projections from the all future climate scenarios.

## RESULTS

### Model performance for potential distribution

The model performance for *A. yasumatsui* was better than random based on 10-fold cross validation, with a mean AUC value of 0.939 ± 0.02 and a TSS value of 0.701 ± 0.11. A "maximum training sensitivity plus specificity" threshold value of 0.112 was obtained from the 10th percentile training presence of the species occurrences. The suitable habitat areas for the pest were reclassified into four levels: <0.112 which indicated unsuitable
**Table 1** Relative contribution of each environmental variables to MaxEnt model.

| Environment variables | Relative contribution |
|---|---|
| Mean temperature of driest quarter (**Bio9**) | 48.1% |
| Altitude (**alt**) | 26.7% |
| Precipitation of coldest quarter (**Bio19**) | 15.2% |
| Mean temperature of wettest quarter (**Bio8**) | 6.8% |
| Mean diurnal range (**Bio2**) | 2.7% |
| Precipitation seasonality (coefficient of variation) (**Bio15**) | 0.5% |

habitat (no risk); 0.112–0.4, which indicated low habitat suitability (low risk); 0.4–0.6, which indicated moderate habitat suitability (medium risk); and >0.6, which indicated high habitat suitability (high risk).

## Important environmental variables

Among the six environmental variables, Mean Temperature of Driest Quarter (Bio9) and Altitude (Alt) had the largest contributions to the distribution model for *A. yasumatsui* (Table 1). These two factors could explain 74.8% of the modeled distribution. The contributions of the other factors, i.e., Precipitation of Coldest Quarter (Bio19), Mean Temperature of Wettest Quarter (Bio8), Mean Diurnal Range (Bio2) and Precipitation Seasonality (Coefficient of Variation) (Bio15) had contributions of the model were 15.2%, 6.8%, 2.7% and 0.5%, respectively, to the model. It appeared thermal conditions and topography were more important than other variables in creating the distribution map for this pest.

## Current invasion pattern

The current distribution pattern of *A. yasumatsui* based on the current environmental variables and occurrence records (including invasive records and native records) is shown in Fig. 1. The current climatic variables in the invasive range of the pest, as predict from MaxEnt, were used to identify potential distribution areas around the world, which were especially concentrated in North America, South America, Europe and Asia. In the USA, Florida, Georgia, Alabama, Louisiana and Texas had high invasion risks under the current climate scenario and areas with moderate invasion probabilities were also found in eastern Mexico. In addition, the Hawaiian Islands were also identified as having high potential distribution areas on the current map. In Europe, moderate-risk areas included France and Italy. In Asia, India, South China, Malaysia, Vietnam and Japan as well as portions of eastern Laos and Indonesia were the areas with the highest risk on the current potential distribution map. In Africa, the potential distribution area will extend to the east of Africa. Based on current climatic variables, the total area of potential suitable habitat for this pest is approximately 46,432,375 km$^2$, of which 2,390,775 km$^2$ (i.e., 5.1% of the total potentially invadable area) has high habitat suitability (high risk), and 5,557,900 km$^2$ (approximately 11.9% of the total potentially invadable area) has moderate habitat suitability (Table 2).

**Table 2 Area with suitability under different climate scenarios.**

| Range | Level | Current (Km²) | RCP2.6-2050 (Km²) | RCP8.5-2050 (Km²) | RCP2.6-2070 (Km²) | RCP8.5-2070 (Km²) |
|---|---|---|---|---|---|---|
| Global | 0.112–0.4 | 38,483,700 | 32,007,300 (−16.8%) | 30,260,450 (−21.3%) | 30,956,000 (−19.5%) | 30,400,550 (−21%) |
| | 0.4–0.6 | 5,557,900 | 5,335,575 (−4%) | 5,136,400 (−7.5%) | 5,247,825 (−5.5%) | 4,929,475 (−11.3%) |
| | 0.6–1 | 2,390,775 | 2,734,675 (14.3%) | 2,820,350 (17.9%) | 2,801,300 (17.1%) | 2,859,175 (19.5%) |
| | 0.112–1 (in total) | 46,432,375 | 40,077,550 (−13.6%) | 38,217,200 (−17.6%) | 39,006,250 (−15.9%) | 38,189,200 (−17.7%) |
| Europe | 0.112–0.4 | 4,148,625 | 6,346,125 (53.9%) | 6,824,375 (64.4%) | 5,927,825 (42.8%) | 8,447,450 (103%) |
| | 0.4–0.6 | 398,425 | 912,600 (129%) | 982,050 (146%) | 932,150 (133%) | 883,600 (121%) |
| | 0.6–1 | 91,850 | 121,550 (32.3%) | 129,925 (41.4%) | 161,150 (75.4%) | 141,775 (54%) |
| | 0.112–1 (in total) | 4,638,900 | 7,380,275 (59.0%) | 7,936,350 (71.0%) | 7,021,125 (51.3%) | 9,472,825 (104%) |
| Asia | 0.112–0.4 | 7,659,250 | 7,112,625 (−7.1%) | 6,267,600 (−18.1%) | 7,195,725 (−6%) | 5,487,525 (−28.3%) |
| | 0.4–0.6 | 1,953,725 | 1,978,075 (1.2%) | 2,015,450 (3.1%) | 1,979,800 (1.3%) | 2,104,150 (7.6%) |
| | 0.6–1 | 715,000 | 1,350,900 (88.9%) | 1,369,925 (91.5%) | 1,316,900 (84.1%) | 1,500,250 (109%) |
| | 0.112–1 (in total) | 10,327,975 | 10,441,600 (1.1%) | 9,652,975 (−6.5%) | 10,492,425 (1.5%) | 9,091,925 (11.9%) |
| North America | 0.112–0.4 | 2,942,525 | 3,351,250 (13.8%) | 3,581,975 (21.7%) | 3,241,450 (10.1%) | 4,309,500 (46.4%) |
| | 0.4–0.6 | 553,475 | 503,525 (−9%) | 462,750 (−16.3%) | 471,000 (14.9%) | 509,700 (−7.9%) |
| | 0.6–1 | 615,325 | 550,250 (−10.5%) | 605,675 (−1.5%) | 638,900 (3.8%) | 501,250 (−18.5%) |
| | 0.112–1 (in total) | 4,111,325 | 4,405,025 (7.1%) | 4,900,400 (19.1%) | 4,351,350 (5.8%) | 5,320,450 (29.4%) |
| Africa | 0.112–0.4 | 6,374,350 | 3,309,550 (−48%) | 2,660,050 (−58.2%) | 3,217,575 (−49.5%) | 1,938,925 (−69.5%) |
| | 0.4–0.6 | 415,775 | 281,375 (−32.3%) | 250,625 (−39.7%) | 281,350 (−32.3%) | 171,700 (−58.7%) |
| | 0.6–1 | 136,800 | 77,475 (−43.3%) | 54,625 (−60.0%) | 74,700 (−45.3%) | 37,800 (−72.3%) |
| | 0.112–1 (in total) | 6,926,925 | 3,668,400 (−47%) | 2,965,300 (−57.1%) | 3,573,625 (−48.4%) | 2,148,425 (−68.9%) |
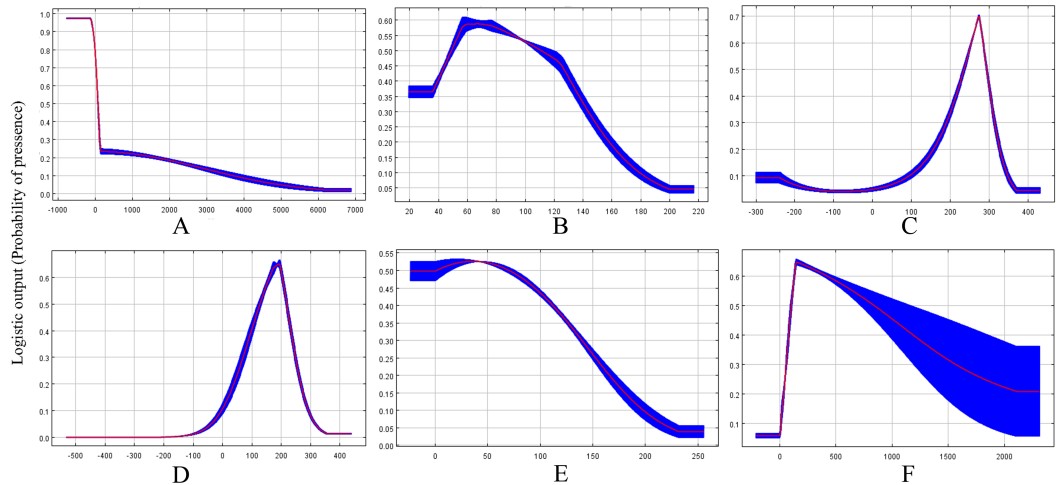

**Figure 2** **Response curves showing the relationships between the probability of presence of CAS and six bioclimatic variables.** Values shown are average over 10 replicate runs: blue margins show ± SD calculated over 10 replicates. (A) Alt; (B) Bio2; (C) Bio8; (D) Bio9; (E) Bio15; (F) Bio19.

Based on the response curves (Figs. 2A–2F), the climatic conditions associated with high habitat suitability were 25–28 °C for Bio8, 15–20 °C for Bio9, 200–400 mm for Bio19. However, both Bio2 and Bio15 shown no relation to high habitat suitability.

## Future invasion risk

The MaxEnt models based on the RCP 2.6 emission scenarios for the potential distribution of *A. yasumatsui* in 2050 are presented in Fig. 3A and Table 2. The potential area of invasion with suitable habitat was 40,077,550 km$^2$, which represents an shrink of 13.6% over the current area of suitable habitat. The area of highly suitable habitat expand to 2,734,675 km$^2$ (an increase of approximately 14.3% over the current highly suitable area).

The potential area of distribution with all suitable habitat will be decrease, but area of highly suitable habitat expand in some area during this period. In Asia (Fig. S2 and Table 2), the model predicts that the amount of suitable habitat in this area will be 10,441,600 km$^2$ based on current climatic variables. Suitable conditions would expand under the RCP 2.6-2050 future climate scenario (Fig. S2 and Table 2), but the areas with high potential invasive ranges were mostly concentrated at low latitudes, where the species will continue to expand to an area of 1,350,900 km$^2$ (an increase of approximately 88.9% over the current highly suitable area) under RCP 8.5-2050. Expansions in the total suitable area and highly suitable area also occurred in Europe (Fig. S3 and Table 2). However, the suitable and highly suitable areas decreased in Africa (Fig. S4 and Table 2) and in North America (Fig. S5 and Table 2) under these climatic scenarios.

Under the RCP 2.6-2070 future climate scenario, MaxEnt predicted a gain in the area of suitable habitat of 39,006,250 km$^2$, approximately 15.9% decrease over the current status (Fig. 3B and Table 2). The highly suitable area might notable increase by 17.1%% under this scenario, expanding to 2,801,300 km$^2$. The restricted range expansion of the suitable habitat area under this climatic scenario also occurred in Italy, the Kingdom of Bhutan,

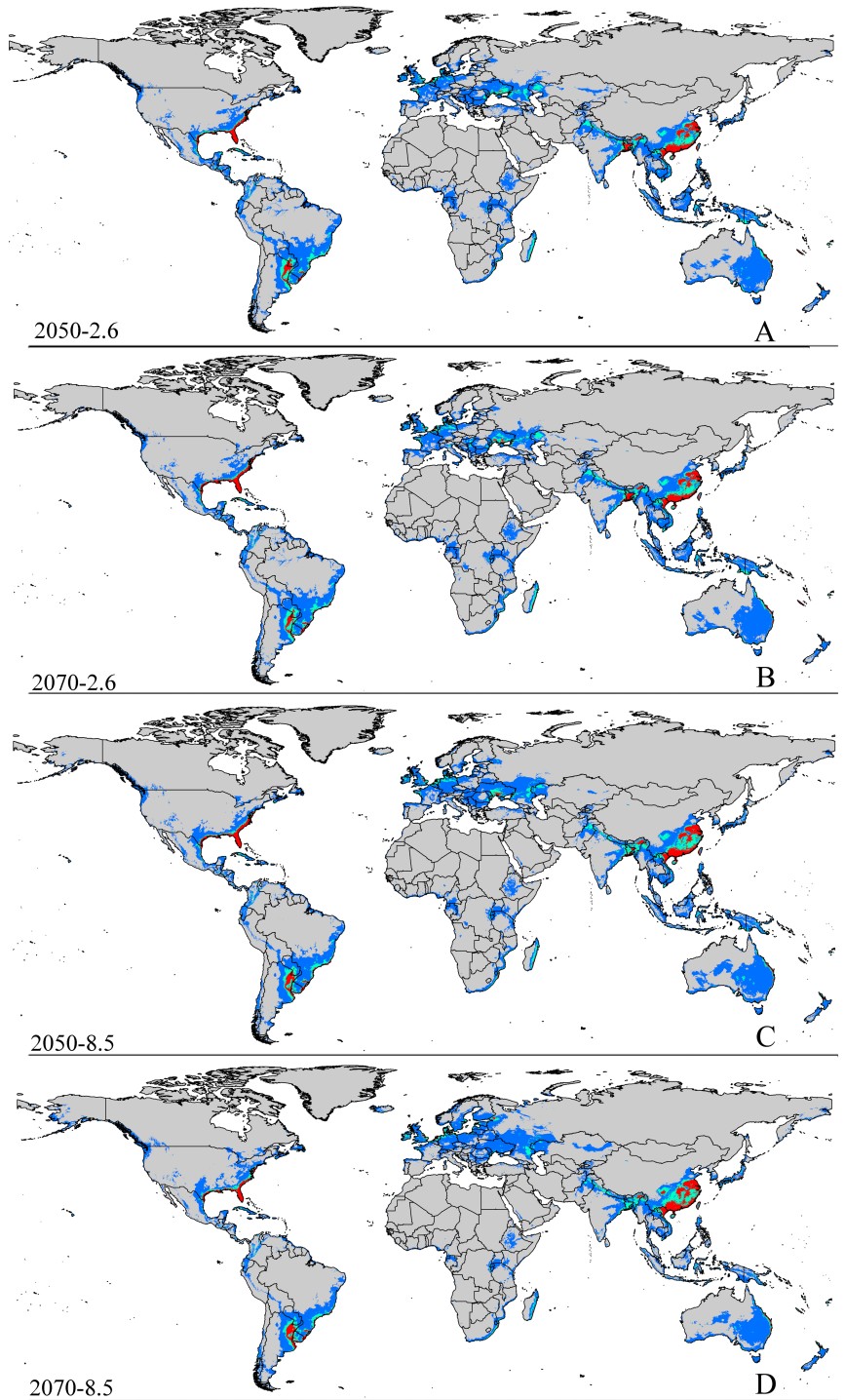

**Figure 3** **Future species distribution models of CAS on global scale under different climate scenarios predicted by MaxEnt.** Gray, unsuitable habitat area; blue, low habitat suitability area; green, moderate habitat suitability area; red, highly habitat suitability area. The base map was created with Natural Earth Dataset (http://www.naturalearthdata.com/). (A) RCP 2050-2.6; (B) RCP 2070-2.6; (C) RCP 2050-8.5; (D) RCP 2070-8.5.

north-west Africa, northern Bangladesh, Southeast India, South China and some regions in Russia (Fig. 3 and Table 2). Moreover, the expansions in the total and highly suitable areas also occurred in Europe (Fig. S3 and Table 2) and Asia (Fig. S2 and Table 2). However, the suitable and highly suitable areas decreased in Africa (Fig. S4 and Table 2) and in North America (Fig. S5 and Table 2) under these climatic scenarios. Under RCP 8.5-2050, the model-predicted area of suitable habitat was 38,217,200 km$^2$, which represents an decrease of approximately 15.9% over the current suitable habitat area (Fig. 3C and Table 2), and the highly suitable area expanded to 2,801,300 km$^2$ (an increase of 17.1% over the current extents). In this scenario, the potential highly potential suitable area in Asia will be increase to 1,369,925 km$^2$ (approximately 91.5% increase over the current area) (Fig. S2 and Table 2) and 129,925 km$^2$ in Europe (approximately 41.4% increase over the current area) (Fig. S3 and Table 2). However, in Africa (Fig. S4 and Table 2) and North America (Fig. S5 and Table 2), the highly suitable area will shrink to 54,625 km$^2$ (i.e., a decrease of about 60% compared to the current extent) and 605,675 km$^2$ (i.e., a decrease of about 1.5% compared to the current extent), respectively.

MaxEnt predicted that the area of suitable habitat under the RCP 8.5-2070 future climate scenario would be 38,189,200 km$^2$, which is actually an decrease approximately 17.7% over the current area (Fig. 3D and Table 2), but the highly suitable area expanded to 2,859,175 km$^2$ (a 19.5% increase). In this scenario (Fig. S2 and Table 2), the potential highly suitable area in Asia will continue to expand to 9,091,925 km$^2$ (an increase of approximately 11.9% over the current extent); in Africa (Fig. S4 and Table 2), the area will shrink to 37,800 km$^2$ (a decrease of approximately 72.3%); and in Europe (Fig. S3 and Table 2), the area will expand to 141,775 km$^2$ (an increase of approximately 54% over the current extent). However, the potential highly suitable area in North America (Fig. S5 and Table 2) will further decrease to 501,250 km$^2$.

In short, the potential distribution of this pest will increase with climate change. Specifically, the area of highly suitable habitat will increase in most areas, but decreases in both the areas of suitable and highly suitable habitats will occur in Africa (Fig. S4 and Table 2) and also in North America (Fig. S5 and Table 2) under the future climatic scenario.

## DISCUSSION

### Change in habitat suitability

In this paper, the potential distribution of the invasive CAS was first simulated using MaxEnt and presence data. The CAS is native to Thailand and has already invaded other continents. The results of the simulation of the potential distribution of this invasive scale insect showed that climate change would affect its worldwide distribution. Our model also suggests that the CAS has a high invasive potential and is capable of invading large portions of the global landmass on several continents, particularly in South China, North India, north-western Africa and some areas of Russia. The occurrence of unsuitable areas on the current distribution map of *A. yasumatsui* suggests that these areas are constrained by inappropriate ecological conditions due to natural enemies or local adaptation, but these
areas are expected to change over time with the increase in climate warming. This pest is already causing serious damage to the mature leaves and trunks of many cycad species (*Normark et al., 2017*) in Southern China, Guam, and Florida, where the pest is invasive. Interestingly, our results suggests that western Australia, northern New Zealand, western Madagascar, southern Brazil, Uruguay and northeastern Argentina have suitable habitat for this pest; however, fortunately, there have not been any reports of its presence at these sites. Thus, strict quarantine measures should be enforced across these regions to prohibit the spread of this pest from its invasive and native ranges. The potential distribution map for this pest provides a reference for developing monitoring strategies to detect future infestations in currently uninfected regions.

However, not all regions showed an expansion of the potential distribution range under climate change. Based on two climate scenarios, the current research showed that climate change will reduce the risk of invasion on some continents, such as Africa and previous studies have also suggested that some invasive species are likely to experience range reductions due to climate change (*Sarma, Munsi & Ananthram, 2015*; *Bradley et al., 2010*; *Roura-Pascual et al., 2004*). Retreat by invasive pests could lead to opportunities for the restoration of currently invaded landscapes, but further research is needed to identify these opportunities and provide sound guidance for ecological restoration.

## Factors influencing invasion

The results of the potential CAS distribution modelling by MaxEnt showed that the Mean Temperature of Driest Quarter is the most important climatic variable defining the current global distribution of this pest. Previous results have demonstrated that CAS eggs hatch to nymphs in 8–12 days when the temperature is 24.5 °C (*Howard et al., 1999*) and adult female would be not laid eggs when temperature at 18 °C and 35 °C (*Cave, Sciacchetano & Diaz, 2009*). Moreover, 25 °C–28 °C has been confirmed to be the most suitable temperature range for the development and reproduction of CAS (*Ravuiwasa et al., 2012*). However, temperatures below 15 °C have been shown to be unsuitable for this insect, as indicated by the high mortality observed in laboratory research (*Teaaro, 2009*). Moreover, some research has suggested there were significant correlations between population growth and temperature variations. Our results confirmed that the mean temperature of driest quarter is the most important factor affecting the model, accounting for 48.1% of the contribution of all climatic variables. Our model suggested that the population of this pest would increase under an mean temperature of 15–20 °C in the driest quarter and a mean temperature of 25–28 °C in the wettest quarter. Therefore, these temperature values represent the most suitable climatic conditions for the population growth of this pest and some currently unsuitable habitats, such as southern China, may be come suitable in the future as currently suitable areas become too hot for CAS development. In contrast, some currently highly suitable habitats, such as North America, may become unsuitable due to global warming. This decrease in habitat suitability in these regions may constraint the spread of CAS outside its current suitable range or limit its development.

Although the potential distribution of CAS under climate change was predicted by MaxEnt, other factors may have limited the accuracy of our results. Only constraints
caused by climate conditions were considered in the current research; however, other factors, such as host-plant availability (*Ning, Wei & Feng, 2017*), interspecific interactions (*Gao & Reitz, 2017*), and dispersal ability (*Guisan & Thuiller, 2005*), also affected the precision of the model; these factors should considered in future research.

## Implications for management strategies

Reasonable management strategies and effective control methods are the key to controlling the spread of CAS and they should be based on a broad consideration of the relationship between invasive species and other factors such as climate change, land use, and human activities (*Pyke et al., 2008*). Based on the results of the current and past studies, one possible method and management strategy for controlling this species is proposed here.

First, identifying the areas experiencing climate change is a critical step (*Pyke et al., 2008*). Species distribution modeling (SDM) is a cost-effective, simply early warning system that enables the identification of potential invasion areas (*Warren, 2012*). Forecasting future areas of invasion by SDM provides an opportunity for governments to prioritize those regions for management and then develop strategies to control pests in these areas. For example, our results suggest that CAS has a highly suitable environment in southern China, which has severely harmed *Cycas revoluta* in Shenzhen City of Guangdong Province (*Yang et al., 2009*) and *Cycas guizhouensis* in Guizhou Province (*Wu, Li & Luo, 2008*); however, unfortunately, no management strategy for controlling this pest is exists in these region. Moreover, some regions, such as western Australia, northern New Zealand, western Madagascar, southern Brazil, Uruguay and north-eastern Argentina, were identified as having suitable habitat for the CAS in our work. Therefore, these areas should formulate strict quarantine measures to prevent invasion by this pest.

Second, many studies have shown that the use of oil (organocide or paraffin-based ultra-fine horticultural oils) on foliage can significantly reduce CAS population size, but several applications may be required (*Hodges, Howard & Buss, 2003*). Conventional insecticides, such as cygon and pyriproxifen (*Weissling, Howard & Hamon, 1999*; *Emshousen, Mannion & Glenn, 2004*; *Howard et al., 1999*), can also control CAS in some instance. However, the chemical control efficiency is low, because the scale can protect itself by hiding between the leaves or stems of cycads (*Hodges, Howard & Buss, 2003*). Therefore, a combination of biological control agents and chemical control can improve management efforts.

Finally, natural enemies are likely another effective way to control CAS. The first biological control agents, a parasitic wasp, *Coccobius fulvus* and a predatory beetle, *Cybocephalus nipponicus*, were released in 1998 (*Cave, 2006*), but a satisfactory level of control has not been achieved. Since then, many natural enemies have been used to control CAS at certain times and over certain ranges; for example, both *Cybocephalus nipponicus* and *C. flavocapitis* are effective biological control agents of CAS in Taiwan Province, China (*Bailey, Chang & Lai, 2011*). Approximately 15,000 *Coccobius fulvus* individuals were released in 14 Florida counties during February–April 2002, and good CAS control results were observed (*Cave, 2006*). Moreover, some new natural enemies have been found to feed on CAS, but their control efficacy needs to be further verified. These biological control agents included *Rhyzobius lophanthae*, *Aphytis lingnanensis* and *Aphytis lepidosaphes*.

In brief, areas that have been invaded by CAS must combine chemical control with natural predators to manage this pest. Additionally, strict quarantine measures should be implemented in areas where this pest has not yet been recorded but where climatic conditions appear to be suitable to CAS, as suggested by our results.

## CONCLUSIONS

Invasive species are considered a major threat to ecosystem functioning and native biodiversity. Climate changes can enhance invasion processes from initial introduction through establishment and spread, and consequently have a profound influence on agricultural systems. The CAS is one of the most serious pests of cycads that has emerged in recent years. This study provided the first potential distribution map based on current and future climate scenarios, and the result showed that the potentially suitable habitat area or CAS is far greater than the current distribution under current climate conditions. The amount of suitable habitats will continue to increase under future climate change scenarios. The present study only provides a reference to the manage this invasive species and develop policies for its control; thus, further research is required.

### Funding

This work was supported by the National Science Foundation Project of China (no. 31301899 and no. 31501876), supported by Natural Science Foundation of Shanxi (no. 201601D021122) and Shanxi Agricultural University of Science and Technology Innovation fund projects (2015YJ03). The funders had no role in study design, data collection and analysis, decision to publish, or preparation of the manuscript.

### Grant Disclosures

The following grant information was disclosed by the authors:
National Science Foundation Project of China: 31301899, 31501876.
Natural Science Foundation of Shanxi: 201601D021122.
Shanxi Agricultural University of Science and Technology Innovation fund projects: 2015YJ03.

### Competing Interests

The authors declare there are no competing interests.

### Author Contributions

- Jiufeng Wei conceived and designed the experiments, performed the experiments, analyzed the data, contributed reagents/materials/analysis tools, prepared figures and/or tables, authored or reviewed drafts of the paper, approved the final draft.
- Qing Zhao conceived and designed the experiments, performed the experiments, prepared figures and/or tables, authored or reviewed drafts of the paper, approved the final draft.

- Wanqing Zhao analyzed the data, approved the final draft.
- Hufang Zhang conceived and designed the experiments, authored or reviewed drafts of the paper, approved the final draft.

## Data Availability

The raw data are included in the Supplemental Files.

## Supplemental Information

Supplemental information for this article can be found online at http://dx.doi.org/10.7717/peerj.4832#supplemental-information.

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
