# Peer review of "Predicting the potential distributions of the invasive cycad scale Aulacaspis yasumatsui (Hemiptera: Diaspididae) under different climate change scenarios and the implications for management"

_PeerJ, doi:10.7717/peerj.4832_

## Round 0.1 · original submission · Major Revisions

I have now received three reviews of your article from three outside readers, appended below. These reviews indicate that while your article potentially presents an important contribution to the field, revisions are necessary before your article could be accepted for publication. Reviewers 1 and 3 had several specific questions about the methods and modelling and offered helpful suggestions on ways to improve the text.

If you are prepared to undertake these revisions, we look forward to receiving your revised manuscript, and ask that you please directly respond to the readers' comments when you submit it.

Reviewer 1 ·

Basic reporting

Of the most urgent calls among biologist is that for native species protection. Invasive species are the most damaging of insults to natural diversity, presumably more destructive than land conversion, as the invasives are only just getting started with a new blight or predator being recognized possibly every day, worldwide. The authors offer new knowledge to project future species distribution of the cycad scale, which is devastating to both horticulture and native cycad species and has made a rapid worldwide assault on cycads. In the last decade scale has created large swaths of cycad-less cycad forest and the need to develop new control methods is paramount to control of this species, with concerted projection efforts meeting only limited success. Thus, it is welcome to see modeling efforts identify and designate the most vulnerable flora needed for protection including quarantine, protective surveys, local population education programs to design quick spotting and eradication planning methods and the design of new biological technology to eliminate the pest. However, the manuscript is not ready for acceptance and unfortunately needs to be rejected due to insufficient preparation, as well as improper bioinformatics follow through.
In addition the manuscript is riddled with spelling and grammatical mistakes such that it makes it difficult to judge the quality of work and to determine what the authors intended to convey.

Below are a list of grammatical/spelling/concept clarification mistakes that I noted at the outset but gave up on listing additional problems as they are far too numerous to extract.

Title
Predicting the potential distributions of the cycad aulacaspis scale Aulacaspis yasumatsui (Hemiptera: Diaspididae) under climatie change scenarios and the implications for management
-remove first appearance of the word aulacaspis
-climate misspelled
-perhaps add the word, “invasive”

Abstract

The very first sentence of the Abstract contains multiple grammatical errors,
“Cycads, an ancient group of gymnosperms, that are almost all threatened or endangered
an are now popular landscape plants.”
-It should state, “Cycads, an ancient group of gymnosperms, nearly all threatened or endangered in the wild
are also popular landscape plants.

There is something not right about this sentence. I have underlined the problematic portion. “A potential risk map of CAS was created by (add by using MaxEnt and using occurrence data under changing climatic conditions.”

“Moreover, this research provide a theoretical reference framework for developing policy for the management and control of this invasive pest.”
-Should be “provides” for proper agreement in the sentence

The model suggested the current invasive risk was mainly constrained
by the annual temperature range (Bio07), mean temperature of coldest quarter (Bio11)
and mean temperature of driest quarter (Bio09).
-The model forumulated through this method suggests that
-Odd to included (Bio7, 11 & 9) in the abstract as it has no meaning to the reader if the abstract is read by itself.

The potential expansions or reductions of distribution ranges were also predicted under different climate change conditions.
-This could be interesting, but which climate change conditions?

Although biotic factor and spread factors were not considered in the current analysis, using climatic
factors to achieve a better understanding of the invasion patterns of this species can help
improve the management of this invasive species and develop policies for its control.
-Again, sentence is rather pithy like the one before it. Which climatic factors? Why not state the findings in the paper?

Introduction

45-46…Invasive species>,> not only threaten.s> native ecosystems through interspecific competition with
native species>,> but also threaten human-managed systems, such as agriculture, human and animal
health, and forestry (Bardley, Wilcove & Oppenheimer, 2000; Simberloff et al., 2003).
-Is “human health” a human managed system?

48…Considerable evidence has shown that climate change can significantly influence the distribution
of invasive species by affecting key physiological characteristics
-What does that mean to “affect key physiological characteristics”?

50-51 …Understanding the change in potential distribution as a result of
climate change is a critical foundation required to manage and control the introduction of alien
species
-change in potential distribution of???”
-I’m not sure what this sentence is saying? The presumption is that climate change will exacerbate the spread of invasive species. What is the foundation referred to in relation to invasive species?

56-57…The oldest cycad fossil record was approximately 265 to 290 million
57 years old (Brenner, Stevenson & Twigg, 2003).
Should state fossil record is (Presuming this fossil still exists)

60…Nevertheless, many cycad species, such as cycad genera, are facing mass extinction due to insect pests
-Should list cycad species as examples. Not clear what “such as cycad genera” is stating


62-63…The cycad aulacaspis scale (CAS), Aulacaspis yasumatsui Takagi (Hemiptera: Diaspididae), is one of the most serious pest of cycads (Giorgi &Vandenberg, 2012) .
-I would argue that this is the most serious pest of cycad

64-65…Studies have suggested that this highly invasive species poses a threat to costly ornamental plants, and this pest not only damages wild cycad populations but also infectes the plants in conservation areas around the world
-I think the point here is that much of the destruction of native habitat has occurred through introductions from horticultural plants
-infects

71 subsequently
90 that appear as if covered in awhite crust
93 At present, at least 8 genera from 3 different families cycads

94-95 …There is an urgent need to control and manage this pest due to its spread rapidly and wide host range among cycads.
rapid spread.

99-101…Therefore, a alternative method is to enforce strict quarantine measures in countries where CAS has not yet beeb introduced by prohibiting importation of cycad plants from infested countries.
-an alternative
-not yet been
-is by prohibiting


102-103 Identifying the current and future potential distribution range is very important for early monitoring, formulating the quarantine measures and managing CAS
-This sentence is choppy and missing parallel phrase structure for smoother reading.

Etc.

Experimental design

Beyond the chronic grammatical nuisances, the work itself is generalistic about the subject without reference to confirmatory data that would help validate the model.
For example, the authors argue that temperature is the most significant environmental factor regulating distribution, but this may not be particularly useful considering that cycads are a pan tropical and subtropical species so one would expect most cycad habitat to be suitable as scale habitat, unless the authors can do a better job clarifying this. A correlation between cycad temperature survival zones would be extremely helpful. The authors suggest that the survivability cutoff for the cycad scale is 15C. However, if one considers the low temperature of the initial site of scale invasion that occurred in North America, for example, in Southern Florida, Miami, where the temperature falls to 10C, typically three times a year, the scale invasion has proceeded unabated. The authors thus may want to consider calibrating their analysis to more accurately reflect actual temperature outcomes. If the authors were inferring that the mean temperature cut off is 15C then this should be clarified.
Another important omission is that the authors do not address the vulnerability among the two different cycad subgroups. There is no effort to distinguish or compare probably survivability outcomes between New vs. Old World cycads. Cycad scale is more damaging to the Cycadales when compared to the Zamiales and this distinction could affect scale range sensitivity.

Validity of the findings

In a glaring error, both Figures 1 and 2 are not labeled as the species of study, Aulacaspis yasumatsui, the cycad scale, but instead, Phenacoccus solenopsis, the cotton mealy bug. So, the reviewer is left to wonder whether the data presented comes from a concurrent project on a different species. Either way, this mistake, among the other listed problems makes this work unsuitable for publication.

Reviewer 2 ·

Basic reporting

The manuscript should be checked for minor grammatical errors.
I suggest to move Figs. 5-8 to supplementary material.
Check if abstract complies with PeerJ standards.

Experimental design

No comment

Validity of the findings

Please make clear in the introduction and conclusions that your model is built to predict the potential distribution range of CAS, but it doesn’t take into account the distribution of hosts and insect’s biology (e.g. relationship with temperature).

Additional comments

L29. Avoid “CAS” in the abstract.
L32. Delete “by MaxEnt and”
L45-46. “such as agriculture, forestry, human and animal health”
L56-57. Please delete.
L57-59. It seems contradictory with the following sentence. Please revise.
L71. “Cycas revoluta”.
L121. You mean “single point”?
L130-131. This statement can be misinterpreted. Please, rephrase.
L137-138. Does this sentence belong to the following paragraph?
L142-145 and Tab.1. Please make clear that these are coded by WorldClim
L154. Break into two sentences.
L167. “highest performing methods” is not clear.
L173-176. It is not clear, please rephrase.
L182. Area Under Curve?
L187-189. “from -1 to +1” and I think it is “where 1 indicates …”

Reviewer 3 ·

Basic reporting

The manuscript entitled “Predicting the potential distributions of the cycad aulacaspis scale Aulacaspis yasumatsui (Hemiptera: Diaspididae) under climate change scenarios and the implications for management” generally conforms to PeerJ standards in terms of formatting norms, figure quality and raw data availability. However, there are several aspects of the manuscript that require improvement, including its content, language and presentation. I highlight some of these aspects in detail below:

Firstly, certain passages of the introduction are too general or broad and would benefit from clarification or further detail. For example, line 48 could be improved by stating which physiological characteristics you are referring to. Also, lines 57-61 are somewhat contradictory – you first say cycads are resistant to pests, but then highlight they are endangered mostly due to pest species. A few other sections to re-evaluate include passages of the discussion (see below).

Second, some lines lack a solid reference to support the statement made. Examples include lines 79-80 and 367-369. On the other hand, some references are incorrectly used. For example, Hijmans et al. 2005 is cites as a reference to the fact that observations are often concentrated in highly accessible or populated areas. While this is true, Hijmans et al. (2005) refer to meteorological observations and not species observations. Several more adequate references exist for this purpose, such as Kadmon et al. 2004. Other passages where references are inadequately cited are highlighted below.

Thirdly, the English language should be improved to ensure that an international audience can clearly understand your text. Some examples where the language could be improved include the abstract and lines 65, 68, 86, 95, 100-101, 105, 119, etc. These lines contain either spelling mistakes or passages that have awkward wording and would benefit from rephrasing. Also, there is a lack of careful formatting throughout the manuscript (double or trailing spaces, types, etc.) and I would encourage the authors to carry out a thorough revision of the manuscript. On top of this, authors also should also carefully revise the technical terminology they use throughout the manuscript. For example, the authors clearly mention that “the current and future potential distributions of CAS were estimated using MaxEnt software based on available occurrence data” (which points towards a species distribution modelling mindset) but they then refer to ecological niche modelling in the discussion (lines 350-354). Reviewing the literature on the subject, and on the emphasis of each approach might help the authors to tackle some of the limitation of this study (for further details on limitations, see below). I would encourage them to check the manuscripts by Warren (2012, 2013), McInery & Etienne (2013) and references therein for further guidance.

Experimental design

The research question is clearly outlined and within the scope of the journal. Its meaningfulness arises from the importance of this invasive pest species as clearly outlined by the authors in the introduction. However, some of the methodological decisions raise some serious questions regarding the robustness of the study and some passages need to be more clearly presented to allow readers a full assessment of the study. I highlight some of the key limitations in more detail below.

Firstly, variable selection is limited to climatic variables and is badly justified (lines 128-131). The authors use two references to justify this selection: one is the authors’ own study (Wei et al. 2017) and the second is Guisan et al. 2013. Interestingly, this last reference is a manuscript that highlights the need to carefully consider species distribution model development for conservation purposes, and emphasizes how factors beyond climate can generate uncertainty in model outcomes. For example, Table 1 of said manuscript emphasizes that “Biotic interactions may play a strong role in determining environmental suitability in novel habitats” when assessing biological invasions. While it is true that topography may not play such an important role in this case, as the authors argue, there is not much justification why other factors such as dispersal limitation (e.g. Sullivan et al. 2012) or biotic interactions as argued by Guisan and colleagues. To be honest, the authors recognize this fact in the discussion (lines 338-343) so I am surprised that a stronger justification for including only climatic variables is provided in the methods. I would encourage the authors to revise this section and provide a better justification for only considering climatic factors for modelling purposes – the known relationship between temperature and CAS development could serve this purpose.

Second, the climatic variables selected for modelling purposes are highly correlated. While the authors recognize that collinearity among predictor variables can affect modelling outcomes, highly correlated variables (i.e. |r|>0.7) were still included in the models. Bio2 and Bio10 for example have r=0.799! I would encourage the authors to revisit their choice of predictor variables or to consider a set of uncorrelated variables representing climate (e.g. from a PCA) for modelling.

Thirdly, only one GCM was selected for the study even though uncertainty associated with climatic model predictions is known to be relevant – Guisan et al. (2013), who the authors cited, highlight this clearly. Providing an account of the uncertainty associated with different future scenarios and climatic models would allow a more nuanced discussion of the results obtained, which will certainly be of more use to decision-makers.

Finally, the modelling procedure requires additional details and justification to be assessed adequately. On the one hand, some aspects of the study are poorly described. For example, authors state that “To define the background area, the minimum convex polygon were implemented by ArcGIS 10.1 (ESRI, Redlands, CA, USA).”. But how were these polygons implemented? Were they defined within each region (i.e. North America, Africa, Asia, etc.) or globally? Obviously, either approach will result in greatly different MCPs for the selection of background points. Also, were background points then sample randomly or through a stratified approach? All of these aspects are relevant to assess the study and allow reproduction of the study. On the other hand, some decisions are poorly justified. For example, the authors stated that default features were used for modelling purposes, but decisions regarding the allowed model complexity can have great impacts on final modelling outcomes. I would encourage the authors to read the paper by Merow et al. (2014) which discusses these issues in details, to consider the limitations and objectives of their own study (sample size and bias, spatial extent, extrapolation vs. interpolation) and to provide a more adequate justification for their modelling decisions.

Validity of the findings

The general findings of the study are in agreement with the results presented, but these are hindered by some of the methodological issues raised above. This ultimately reflects on the discussion provided, and there are two sections where that is particularly evident.

In the first section, the authors state that “The occurrence of unsuitable areas on the current distribution map of A. yasumatsui suggests that these areas were constrainted by inappropriate ecological conditions as a result of natural enemies, ecological or geographical barriers to spread and local adaption” (lines 296-298). These lines seem to imply that discrepancies between the ‘known’ distribution of the species exist and that they may be driven by the factors mentioned. Interestingly, many of these factors are of utmost importance to modelling exercises but were not considered in this study. Hence, it seems plausible that discrepancies may be driven by an over-simplistic modelling approach or simply by biases in the dataset used for model calibration – the fact that more locations are known outside than inside the species natural range suggests this. Ultimately, the current justification seems rather simplistic and it would be positive if the authors could provide a much more nuanced discussion on the observed differences and why they may be arising.

In the second, the authors state that “the lowest temperature of 15℃ proved to be unsuitable for this insect, as indicated by the high mortality observed in laboratory research (Teaaro 2009)” (lines 324-326). A few lines later (329-331), they mention that “Our model suggested an increase in the population of this pest with an annual temperature range of 1-8 ℃and a mean temperature of 14-20 ℃ in the coldest quarter”. A detailed look at Figure 3 actually shows that the peak suitability identified by the model regarding the mean temperature of the coldest quarter is around 15ºC, which proved to be unsuitable for the insects in the laboratory. Hence, model results seem to be in clear disagreement with the known ecology of the species and suggests one of two things: either other factors are allowing the species to settle in unsuitable conditions, or the models are not actually representing the climatic tolerance of the species. In either case, a much more detailed and nuanced discussion is necessary here to provide insights on why this may be. My experience, from looking at model response curves, suggests that models may be overfitting the data and with such a limited and biased dataset, this can be an issue.

Having both of these aspects in mind, I would encourage the authors to critically re-evaluate their study and to provide a better discussion of potential limitations and disagreements between their results and existing knowledge of the species’ ecology. This will be of great help for any interested readers, particularly if results are to inform policy and monitoring as the authors emphasize.

Additional comments

I provide here the full details to some of the references mentioned above:
Kadmon, R., Farber, O. & Danin, A. (2004) Effect of road-side bias on the accuracy of predictive maps produced by bioclimatic models. Ecological Applications, 14, 401–413.
McInerny, G. J., & Etienne, R. S. (2013). ‘Niche’or ‘distribution’modelling? A response to Warren. Trends in ecology & evolution, 28, 191-192.
Sullivan, M. J. P., Davies, R. G., Reino, L. & Franco, A. M. A. (2012), Using dispersal information to model the species–environment relationship of spreading non-native species. Methods in Ecology and Evolution, 3, 870–879.
Warren, D. L. (2012). In defense of ‘niche modeling’. Trends in ecology & evolution, 27, 497-500.
Warren, D. L. (2013). Niche modeling’: that uncomfortable sensation means it's working. A reply to McInerny and Etienne. Trends in ecology & evolution, 28, 193-194.

---

## Round 0.2 · Major Revisions

I appreciate your effort reviewing the manuscript, incorporating main issues identified in the previous version. Nevertheless, some inconsistencies and questions remain even in the current version of the manuscript. I feel you could address properly those issues because the potential interest of the study in the field. If you are prepared to undertake these revisions, we look forward to receiving your revised manuscript, and ask that you please directly respond to the reviewers comments when you submit it.

Again, thanks for your interest in PeerJ.

Best regards,

Salva

Reviewer 2 ·

Basic reporting

English must be checked by a native speaker or a professional editing service.

Experimental design

I think you can use the information reported here:

Cave, R. D., Sciacchetano, C., & Diaz, R. (2009). Temperature-dependent development of the cycad aulacaspis scale, Aulacaspis yasumatsui (Hemiptera: Diaspididae). Florida Entomologist, 92(4), 578-581.

to improve the model and the discussions about the validity of the model.

Validity of the findings

No comment.

Additional comments

L37. Report the year.
L429. There is no need to repeat "Cycad aulacaspis scale"
Table 1. Improve the caption, explaining what the acronyms on the first row mean.
Figure 1. Improve the caption, explaining what the map colors mean (grey, yellow, ...)
Figure 2. Use a thicker line for average, and a lighter blue for SD band.

Reviewer 3 ·

Basic reporting

The revised version of the manuscript entitled “Predicting the potential distributions of the cycad aulacaspis scale Aulacaspis yasumatsui (Hemiptera: Diaspididae) under climate change scenarios and the implications for management”, like the original submission, conforms to PeerJ standards in terms of formatting norms, figure quality and raw data availability. I acknowledge the efforts made by the authors in terms of addressing reviewer’s comments and as a result, some aspects of the manuscript have clearly improved, including language and presentation. Nevertheless, some typos and grammatical errors remain in the manuscript. For example, in line 101 “Cycas.taitungensis” should read “Cycas taitungensis”. Similar issues occur with references; in line 121, Jimenez-valverde et al., 2011 should read Jimenez-Valverde et al., 2011 and in the full reference provided at the end of the manuscript, Lobo should also be capitalized. A thorough and careful revision of the text should be carried out by the authors to improve these aspects.

Experimental design

As part of the efforts made by the authors to consider and include reviewer’s suggestions, some methodological aspects of the manuscript were also improved. For example, the evaluation of model parameters and features was an important and welcome addition. Still, while the authors state that the regularization parameter was also evaluated, there is no evidence of how this was carried out and which values were finally selected. Absence of this information prevents a deeper assessment of the revised model by readers and reviewers, and further details should be provided.
On the other hand, some of the changes carried out by the authors raise serious questions regarding their understanding of the methodology and in the choice of modelling decisions. One of the issues pertains to the choice of variables; the authors now state that they performed a principal component analysis (PCA) to remove highly correlated variables (r≤∣0.8∣). However, upon inspection of Table S3, it is evident that the variables selected are actually highly correlated between them, as the authors included in the study those variables that relate most to each PCA axis. For example, it is easy to understand that variables representing precipitation of the driest month (Bio14) and precipitation of the driest quarter (Bio17) will be highly correlated as they represent essentially the same climatic reality. PCA analysis is often used in species distribution modelling due to the fact that the resulting axis are orthogonal and thus represent different dimensions of the dataset. Hence, either the authors use the PCA to transform their environmental variables in this manner, or I would advise for using more standard correlation analyses to inspect which variables to keep.
Another aspect that raises serious concerns is the inclusion of human footprint in the analysis. Firstly, the inclusion of both topography and human footprint in the models remain unjustified, as I highlighted in my previous revision. My initial comment was not in criticising the exclusion of such variables, rather it was arguing that a strong justification for the environmental variables selected in the model effort is needed as it is well known that models based on variables known to affect the species distribution perform better than models using randomly selected environmental factors (see for example Synes, N. W., & Osborne, P. E. (2011). Choice of predictor variables as a source of uncertainty in continental-scale species distribution modelling under climate change. Global Ecology and Biogeography, 20(6), 904-914. doi:10.1111/j.1466-8238.2010.00635.x). Secondly, even though human footprint can be an important factor in the ecology of invasive species, it also tends to be highly correlated with the distribution of observations available for modelling purposes. Hence, it should be no surprise that the model identifies this variable as the most explanatory driver of the species distribution – mostly because it affects the distribution of observations and not necessarily because of its ecological relevance. If the authors choose to include this variable in their models, they must also account for data prevalence and distribution across the world (see for example Kramer‐Schadt, Stephanie, et al. "The importance of correcting for sampling bias in MaxEnt species distribution models." Diversity and Distributions 19.11 (2013): 1366-1379, although many other relevant references are available).
Ultimately, I do not think the model present is fit for purpose (considering the objectives proposed) and would encourage the authors to once again revise their objectives and methodological decisions.

Validity of the findings

Bearing in mind the above comments regarding the methodology, I do not believe the findings are in line with the proposed objectives of the study and would encourage the authors to once again critically evaluate their study. A thorough re-assessment of modelling decisions and a justification for their choices should be provided.

---

## Round 0.3 · Minor Revisions

Dear Jiufeng,

I think changes carried out have improved substantially your manuscript and are almost ready to be accepted. However, some minor changes need your attention. Please, carefully review the manuscript in order to correct some typos and unclear sentences. We look forward to receiving your revised manuscript then.

Again, thanks for your interest in PeerJ.

Best regards,

Salva

Reviewer 3 ·

Basic reporting

Most of the aspects raised in the last round of review have now been addressed and as a result the manuscript has somewhat improved.

Experimental design

No comment

Validity of the findings

No comment

Additional comments

A few typos and unclear statements remain and should be addressed accordingly before the manuscript is acceptable for publication. Examples of typos and or unclear statements feature in lines 42-45, 307-308, 329 and 403, among others. I strongly recommend that the authors carry out a detailed review of the manuscript to avoid such isses.

---

## Round 0.4 · accepted · Accept

I appreciate your effort reviewing the ms. I believe this new version of your ms is ready to be accepted. Congratulations.

Best regards,

Salva

#